# An Improved CNN for Polarization Direction Measurement

Hao Han [1,2], Jin Liu [1,2], Wei Wang [1,2,*], Chao Gao [1,2] and Jianhua Shi [1,2]

1 College of Advanced Interdisciplinary Studies, National University of Defense Technology, Changsha 410073, China; hanhao.20@nudt.edu.cn (H.H.); liujin1344@nudt.edu.cn (J.L.); gaochao14@nudt.edu.cn (C.G.); gexin7651@nudt.edu.cn (J.S.)
2 Nanhu Laser Laboratory, National University of Defense Technology, Changsha 410073, China
* Correspondence: wangwei8610@nudt.edu.cn

**Abstract:** Spatially polarization modulation has been proven to be an efficient and simple method for polarization measurement. Since the polarization information is encoded in the intensity distribution of the modulated light, the task of polarization measurement can be treated as the image processing problem, while the pattern of the light is captured by a camera. However, classical image processing methods could not meet the increasing demand of practical applications due to their poor computational efficiency. To address this issue, in this paper, an improved Convolutional Neural Network is proposed to extract the Stokes parameters of the light from the irradiance image. In our algorithm, residual blocks are adopted and different layers are connected to ensure that the underlying features include more details of the image. Furthermore, refined residual block and Global Average Pooling are introduced to avoid overfitting issues and gradient vanishing problems. Finally, our algorithm is tested on massive synthetic and real data, while the mean square error (MSE) between the extracted values and the true values of the normalized Stokes parameters is counted. Compared to VGG and FAM, the experimental results demonstrate that our algorithm has outstanding performance.

**Keywords:** polarization direction measurement; CNN; global average pooling



## 1. Introduction

Polarization measurement is important in many applications, such as in sky polarized light navigation [1], remote sensing [2–4], the food industry [5–8] and ellipsometry [9–11]. Generally, four methods, including interferometric polarimeter [12,13], temporally modulated polarimeter [14,15], division-of-amplitude polarimeter [16,17] and spatially modulated polarimeter [18–21], are well accepted to solve this problem. In a interferometric polarimeter, the coherent optical paths are constructed, and the polarization information can be calculated from the interference pattern. The interferometric polarimeter is robust and stable, however, the analyzing process of the interference pattern is usually complicated. In a temporally modulated polarimeter, the rotatable or active optical elements are utilized to modulate the incident polarized light in a time sequential, and the polarization state of the incident beam can be obtained by analyzing the time varying intensity signal. The temporally modulated polarimeter is simple and easy to implement, however, its measurement speed is limited, and it is sensitive to the power and wavelength fluctuation of the light sources. In a division-of-amplitude polarimeter, the incident beams are analyzed by several channels with different polarization optics, and the polarization information can be obtained in a single shot. The division-of-amplitude polarimeter is competent for real-time monitoring, but its configuration is usually complicated to adjust. To conquer these issues, some researchers developed a fourth method, i.e., spatially modulated polarimeter. In a spatially modulated polarimeter, the spatially modulated polarization optics, such as micro polarizer arrays, polarization grating, azimuthal or radial polarizers, are utilized to modulate the intensity in the spatial domain, the polarization information of the incident beam can be obtained by processing and analyzing the spatial modulated intensity image. The

spatially modulated polarimeter can achieve the polarization measurement in a compact, rapid and stable way. It is not sensitive to changes in the power and wavelength of the light. However, as the core devices, the spatial modulation devices are difficult to deploy.

Fortunately, a vortex retarder based spatial polarization modulated polarimetry method is proposed [1,11,18]. The vortex retarder is a special wave plate, and it has a constant retardance across the clear aperture, but its fast axis rotates continuously along the azimuth, so it can convert an ordinary polarized light into a vectorial optical beam [18]. Then, the polarization information is included in the light intensity distribution while the vector polarized light field is detected by a polarizer. Compared to other spatial modulation methods, this method has the advantages of stable performance, low wavelength sensitivity, good temperature stability, high modulation quality and low cost. In this method, the polarization information can be extracted using image processing when the pattern of the light is captured by a camera. Consequently, the accuracy of the polarization measurement is determined by the performance of the image processing algorithms.

Recently, image processing algorithms are divided into two categories: traditional methods and machine learning. In traditional methods, the design of feature extractors relies on the designers' professional knowledge. Furthermore, the methods usually need complex parameter tuning processes. To the best of our knowledge, two papers use tradition methods to calculate the polarization state from the irradiance image. In reference [21], an image correlation operation is proposed to extract the polarization direction from the hour-glass-shaped intensity image. However, the measurement accuracy is decided in the step of correlation operation, and numerous calculations need to be performed to ensure a high accuracy, which is very time consuming. What is more, the method can only obtain the polarization direction, and other polarization information, such as ellipticity and polarization handiness are lost. To obtain the Stokes parameters of the polarized light, a Fourier analysis method (FAM) is proposed [18]. In this method, a series of Randon transformations is performed to obtain the modulation curve of the intensity image, and the Stokes parameters of the incident light can be measured by Fourier analysis of the modulation curve. However, the computational efficiency of the Fourier analysis method is rather poor due to numerous redundant calculations in the Randon transformation. Due to a series of advantages, such as excellent performance, better generalization, end-to-end training and no need for complex parameter tuning, the machine learning method has been widely used in image processing. As the most important branch of machine learning, deep learning performs well and has been widely used in image processing. In reference [22], more than 300 research contributions on deep learning techniques for object detection are introduced. More than 100 deep-learning-based methods have been proposed for image segmentation [23]. Some researchers are devoted to achieving image registration depending on deep learning [24]. Zhao et al. [25] designed SpikeSR-Net to super–resolve a high-resolution image sequence from the low-resolution binary spike streams. In particular, in reference [26], a Convolutional Neural Network (CNN) based on VGGNet architecture was trained to obtain the polarization states of light using a single shot of intensity image. Though it has similar accuracy to FAM, it is much less time consuming.

In this paper, a deep learning technique is also adopted to extract polarization information from the irradiance image due to its outstanding performance. In this paper, an efficient deep-learning-based image processing algorithm, named ResNet-GAP, is proposed to extract the polarization direction from the irradiance image of the modulated input light. To prevent overfitting of the network, global average pooling [27] (GAP) is introduced while ResNet [28] is adopted as the main architecture of our network. Furthermore, the residual block is refined in order to extract image features better and avoid gradient vanishing. In addition, the originally full connection layer is divided into two layers, including a FC layer and a ReLU activation function.

The main work of our paper is as follows: Section 2 introduces the theoretical and experimental investigation, and Section 3 describes our experiments and analyzes the results. Section 4 summarizes the main work of this paper and introduces our future work.

## 2. Theoretical and Experimental Investigation

The schematic of our experiment includes two stages: irradiance image generation and image processing. In the first stage, a spatially modulated scheme using a vortex retarder is built, and the irradiance image including the polarization state of the light is captured by a camera. Then, in the second stage, an improved CNN is proposed to extract the polarization information from the irradiance image.

### 2.1. Irradiance Image Generation Stage

In 1852, Stokes proposed that the polarization state of the light wave can be represented by four real number parameters, which are called Stokes parameters. Generally, the Stokes parameters can be written as a column vector of one order, i.e., $S = \begin{bmatrix} S_0 & S_1 & S_2 & S_3 \end{bmatrix}^T$. For a certain optical element or an optical system, the relationship between outgoing light and the incident light can be represented by

$$S_{out} = M \cdot S_{in} \tag{1}$$

$S_{out}$ and $S_{in}$ are the Stokes vector of outgoing and incident light. $M$ is the Muller matrix of the optical system.

As shown in Figure 1, an integrating sphere (IS) and a positive lens are utilized to generate a uniform and collimated natural light field. Then, the incident light with different polarization states are captured while altering the azimuth angles of the transmission axis of the polarizer and the fast axis of the wave plate. Subsequently, the polarized light is modulated by a retarder and another analyzer. For the zero-order vortex half-wave retarder (VHWR), with the initial fast axis oriented along $0^\circ$, the Equation (1) can be rewritten as

$$S_{out} = \begin{bmatrix} 1 & 0 & 0 & 0 \\ 0 & \cos 2\varphi & \sin 2\varphi & 0 \\ 0 & \sin 2\varphi & -\cos 2\varphi & 0 \\ 0 & 0 & 0 & -1 \end{bmatrix} S_{in} \tag{2}$$

where $\varphi$ is the azimuth angle. Consequently, when the transmission axis of the analyzer is oriented at $0^\circ$, the light intensity can be denoted by

$$I(\varphi) \propto S_{out0} = S_{in0} + S_{in1} \cos 2\varphi + S_{in2} \sin 2\varphi \tag{3}$$

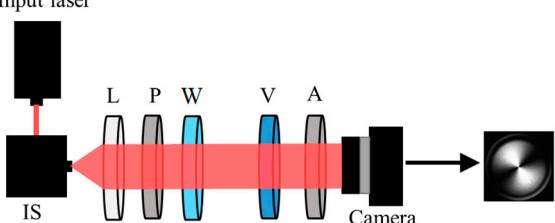

**Figure 1.** The experiment system. IS: integrating sphere; L: Lens; P: polarizer; W: wave plate; V: vortex retarder; A: analyzer.

Similarly, if the incident light is modulated by a zero-order vortex quarter-wave retarder (VQWR), while the initial fast axis is oriented along $0^\circ$, the light intensity is

$$I(\varphi) \propto S_{out0} = S_{in0} + \frac{1}{2}S_{in1} + \frac{1}{2}S_{in1}\cos 2\varphi + \frac{1}{2}S_{in2}\sin 2\varphi - \frac{1}{2}S_{in3}\sin\varphi \tag{4}$$

Equations (2) and (3) point out that, utilizing the optical system shown in Figure 1, the Stokes parameters are encoded in the intensity of the modulated light. In other words, the problem of polarization measurement can be treated as an image processing problem to extract Stokes parameters from the irradiance images.

*2.2. ResNet-GAP*

However, experiments show that the performance of the classical ResNet50 did not meet our expectation in evaluating the Stokes parameters of the light intensity image. Hence, we modify the ResNet50 architecture to make it suitable for our Stokes parameters evaluation problem. The full specification of our modified network, which we called ResNet-GAP, appears in Table 1. The main architecture of our ResNet-GAP is similar to ResNet50, however, there are some improvements.

**Table 1.** The architecture of ResNet-GAP.

| Layer Name | Output Size | Layers |
|:---:|:---:|:---:|
| conv 1 | $112 \times 112$ | $3 \times 3$, stride 2<br>$3 \times 3$, stride 1<br>$3 \times 3$, stride 1 |
| conv2_x | $56 \times 56$ | $3 \times 3$ max pool, stride 2<br>$\begin{bmatrix} 1 \times 1, 64 \\ 3 \times 3, 64 \\ 1 \times 1, 256 \end{bmatrix} \times 3$ |
| conv3_x | $28 \times 28$ | $\begin{bmatrix} 1 \times 1, 128 \\ 3 \times 3, 128 \\ 1 \times 1, 512 \end{bmatrix} \times 4$ |
| conv4_x | $14 \times 14$ | $\begin{bmatrix} 1 \times 1, 256 \\ 3 \times 3, 256 \\ 1 \times 1, 1024 \end{bmatrix} \times 6$ |
| conv5_x | $7 \times 7$ | $\begin{bmatrix} 1 \times 1, 512 \\ 3 \times 3, 512 \\ 1 \times 1, 2048 \end{bmatrix} \times 3$ |
| | $1 \times 1$ | GlobalAveragePooling2D; 1024-d FC, ReLU;<br>dropout; L-d FC, Sigmoid |

In conv1 layer, we choose to replace the $7 \times 7$ convolution with multiple layers of smaller convolution. Convolutions with larger spatial filters (e.g., $7 \times 7$) tend to be time consuming, while convolutions with smaller spatial filters (e.g., $3 \times 3$) tend to be much easier in terms of computation. Hence, we replace the $7 \times 7$ convolution with a concatenation of three layers of $3 \times 3$ convolution (one with stride 2 and two with stride 1). This setup clearly reduces the parameter count and also increases the network depth to maximize the utilization of the network capacity and complexity.

In conv5_x, the first layer is the down sampling module, which, in fact, contains Path A and Path B as in Figure 2. Originally in ResNet50, Path A first completes the channel contraction through $1 \times 1$ convolution with stride 2 to realize down sampling, then a $3 \times 3$ convolution follows, which keep the number of channels unchanged; the main purpose is to extract features and, at the last step, expand the number of channels through a $1 \times 1$ convolution, while Path B is convolved through a $1 \times 1$ with stride 2 for down sampling. Here, in Path A, we move the down sampling process into the $3 \times 3$ convolution step to avoid information loss in the beginning step as a result of the $1 \times 1$ convolution with stride 2, while, in Path B, we use average pooling instead of down sampling.

In the pooling layer, global average pooling (GAP) is used in our modified architecture, which effectively prevents overfitting of the network, strengthens the consistency of feature maps and labels and speeds up network convergence. On the other hand, originally in the ResNet50, there is only one fully connected (FC) layer. Here, we divide it into two FC layers. In the first FC layer, the number of the channels is set to be 1024, followed by a ReLU activation function. Then, dropout operation is employed, which can randomly inactivate some of the nodes to be 0 with probability 0.5 to avoid overfitting. Finally, the channels of the pooling layer are changed from 1000 to L, which is the number of the Stokes

parameters. In the second FC layer, the activation function is changed from Softmax to Sigmoid. Though the Softmax activation function is widely used in multi-label image classification, transforming the output of the model into a probability distribution, since the sum of the probabilities of all categories equals 1, if the probability value of the model output is very small or large, problems may occur such as numerical overflow or gradient vanishing when choosing Softmax activation function, which may reduce the effectiveness of the network. Meanwhile, it is unlikely to meet such a problem when choosing the Sigmoid activation function, since the output range of the Sigmoid function is between 0 and 1, which means that the Sigmoid activation function is more stable in the training process compared with the Softmax activation function.

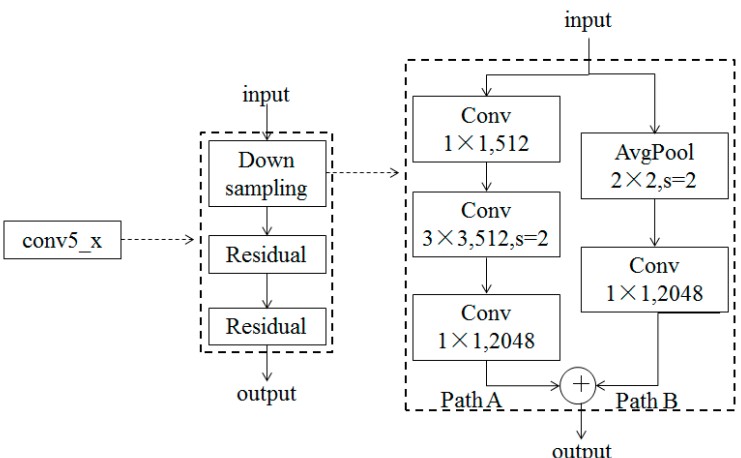

**Figure 2.** The structure of conv5_x.

## 3. Experiments and Results

In the subsequent experiments, we employ the Tensorflow framework under Python to construct, configure, train and test the network. And the network is tested in the same hardware configuration. The network is designed following the configuration to train parameters: the Adam optimizer and the MSE function are chosen as the optimizer and the loss function. The custom evaluation function is ownAccuracy. The batch size is 128, i.e., the number of images fed into the network at each time is 128. The accuracy of the network was assessed using the Mean Squared Error (MSE) metric. In order to completely evaluate the performance, in this paper, the algorithms are tested on images when light is modulated by the VHWR and VQWR, respectively. A vast quantity of experimental data are generated by Matlab for network training. Then, the trained network is tested using synthetic and real images, respectively. To evaluate the robustness of the algorithms to noise, Gaussian noise, in which the mean value is 0 and variance values range from 0 to 0.01 in steps o 0.001, is added to the synthetic images. Two state-of-the-art methods (i.e., VGG and FAM) are adopted for comparison.

### 3.1. Vortex Half-Wave Retarder

3.1.1. Noise-Free Data

Train

We generate noise-free data to train the network. In the training stage, we generate 50,000 perfect, noise-free images. Out of these, 40,000 are randomly chosen for training and 10,000 for validation. As the Stokes parameters are set randomly, 40,000 synthetic images contain enough polarization states of light waves. Theoretically, the trained network can study the characteristics of the training data well.

In the training stage, the size of the batch is set to be 128. MSE of 50 epochs are shown in Figure 3. It can be found that the MSE of the training set is smaller than the validation set. Additionally, the MSE on both sets tend to decrease gradually and converge.

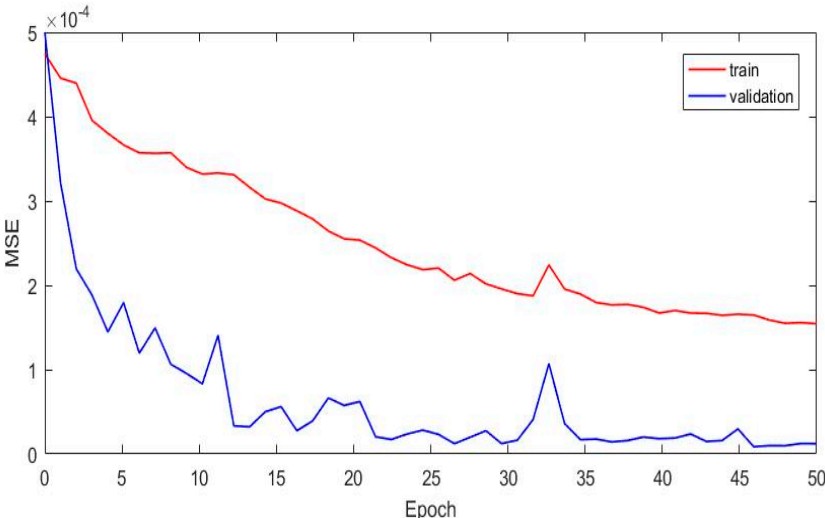

**Figure 3.** The MSE on the noise-free images corresponding to the training set and the validation set for VHWR.

Test

Based on the results obtained from the above training, the model with the best performance in the training process was selected as the test model. We simulate 11 test sets, each containing 1000 frames of images with zero mean Gaussian noise and variance ranging from 0 to 0.01 in steps of 0.001. We select one image from each group as an example, as shown in Figure 4. The MSE of S1 estimated by ResNet-GAP, VGG and FAM is shown in Figure 5. The MSE of S2 is shown in Figure 6. The training data shows that the MSE of the test set is extremely low, and it can even reach as low as $1 \times 10^{-6}$. With an increase in the variance of Gaussian noise, the MSE of S1 and S2 also increases gradually, indicating that the model is sensitive to noise. From Figures 5 and 6, it is evident that the MSE of ResNet-GAP has the slowest growth rate with an increase in noise. That is to say, ResNet-GAP is more robust to noise than VGG and FAM.

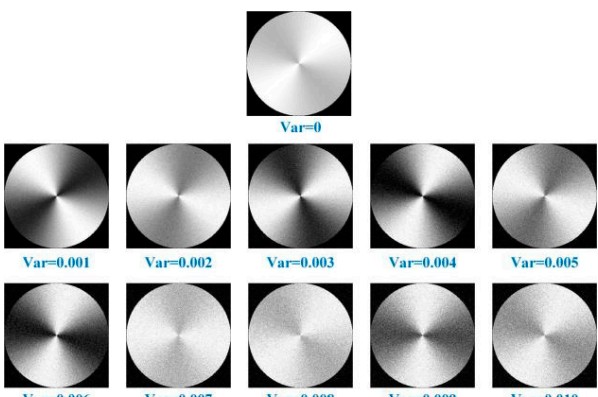

**Figure 4.** Images disturbed by different levels of noise while the light is modulated by a VHWR.

Furthermore, the performances of different algorithms are tested on real data. A total of 37 real images (shown in Figure 7) are captured by our experimental system, in which the transmission axis of the polarizer was fixed at 20°, and the fast axis of the VHWR was rotated from 0° to 180° with a step of 5°. The results for S1 and S2 are shown in Figure 8. They demonstrate the perfect performance of the algorithms. For ResNet-GAP, the maximum absolute error of S1 is 0.0428, and the average value is 0.0035. For S2, the maximum absolute error is 0.0574, and the average error is 0.0014. The MSE of three algorithms are given in Table 2. This illustrates that the ResNet-GAP network achieved an MSE of $1 \times 10^{-4}$ when tested on real images, which is marginally larger

than the results on the noise-free simulated image. This difference is due to noise in the real image.

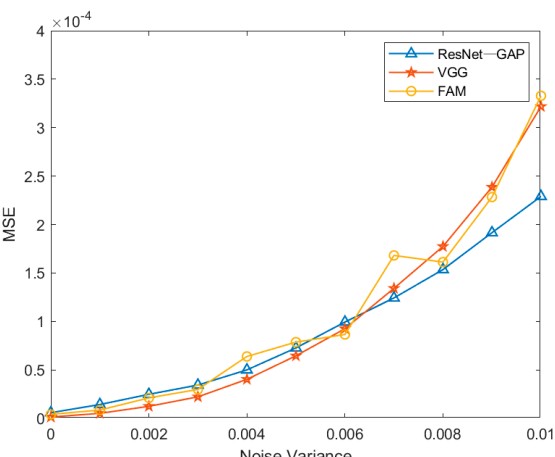

**Figure 5.** The MSE of S1 with respect to noise while the light is modulated by a VHWR.

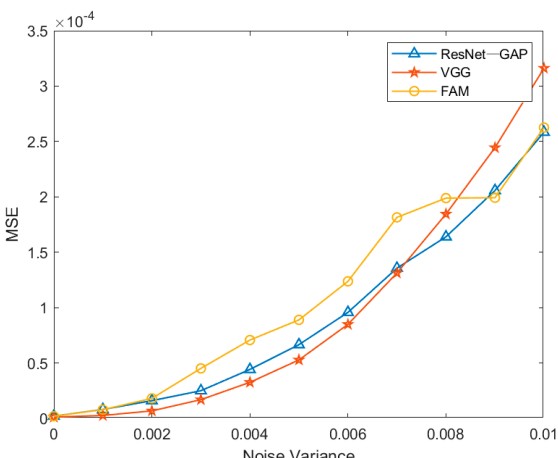

**Figure 6.** The MSE of S2 to noise while the light is modulated by a VHWR.

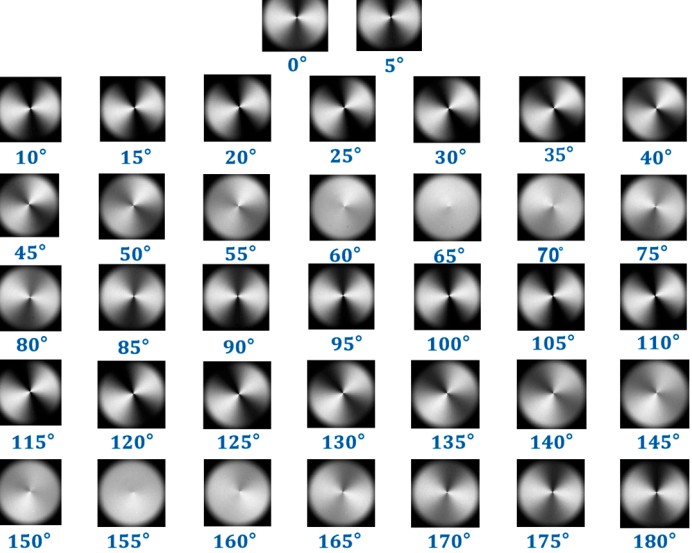

**Figure 7.** The real images when the fast axis of the VHWR is rotated.

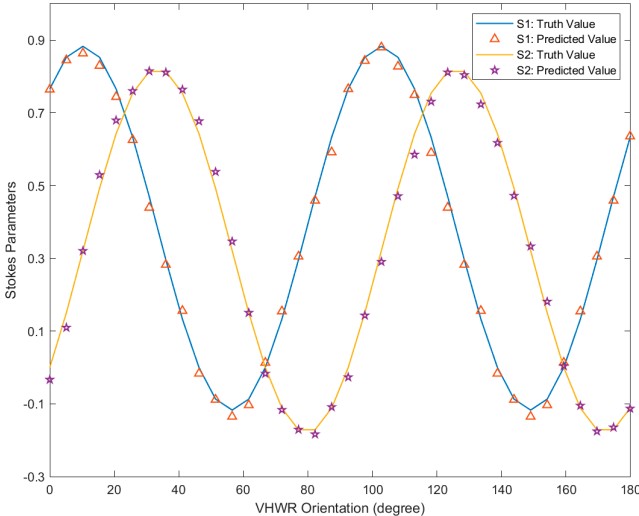

**Figure 8.** The Stokes parameters real values and their evaluated values by ResNet-GAP.

**Table 2.** Performance of algorithms on real images for VHWR.

| Stokes Parameters | Algorithm | MSE |
|---|---|---|
| S1 | ResNet-GAP | 0.000385 |
| S1 | FAM | 0.001623 |
| S1 | VGG | 0.000741 |
| S2 | ResNet-GAP | 0.000506 |
| S2 | FAM | 0.000932 |
| S2 | VGG | 0.000611 |

The truth values of S1 and S2 are plotted against the predicted values of the three algorithms, as shown in Figures 9 and 10. They indicate that ResNet-GAP has the closest predicted value to the truth value and performs better than VGG and FAM.

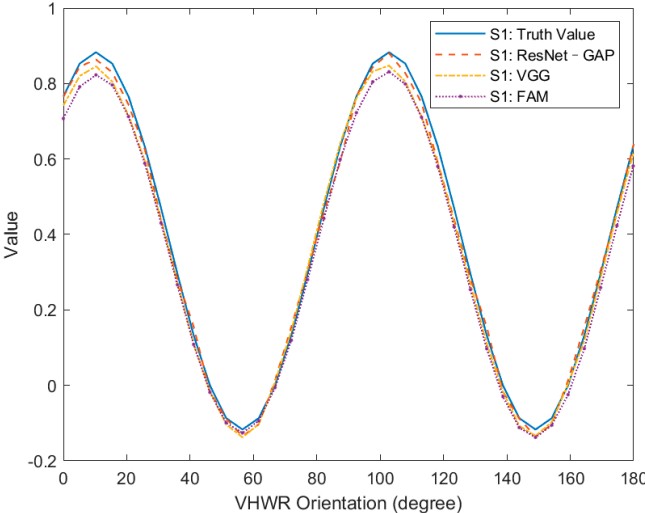

**Figure 9.** The values of S1 to different orientations of the fast axis for VHWR.

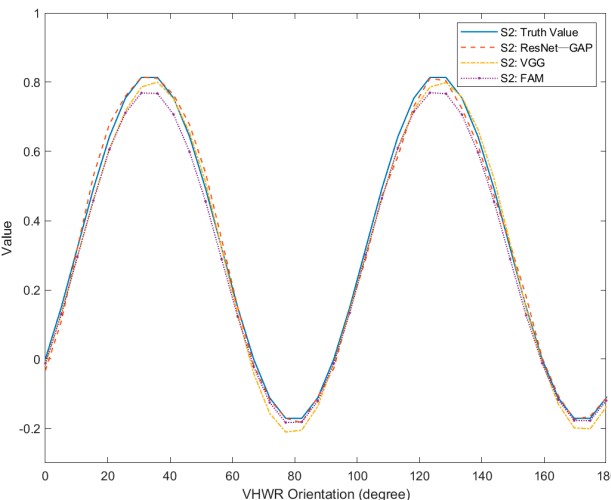

**Figure 10.** The values of S2 to different orientations of the fast axis for VHWR.

When 37 real images were tested, ResNet-GAP consumed only 0.01443 s, in comparison to VGG and FAM, which took 0.003694 s and 28.763 s, respectively. Obviously, the ResNet-GAP network is much faster than FAM and VGG. It is more suitable for real-time processing.

### 3.1.2. Noisy Data

Since noise is usually unavoidable in reality, we selected Gaussian noise images with variance 0.01 as the training and validation data sets. The numbers of images for training and validation are 40,000 and 10,000, respectively. Other parameters are kept the same as the noise-free case described above. The MSE of different epochs are shown in Figure 11. It can be seen that the MSE of the training data set is much less than that of the validation data set. The MSE of the training and validation sets converge quickly, approximately in the magnitude of $1 \times 10^{-3}$.

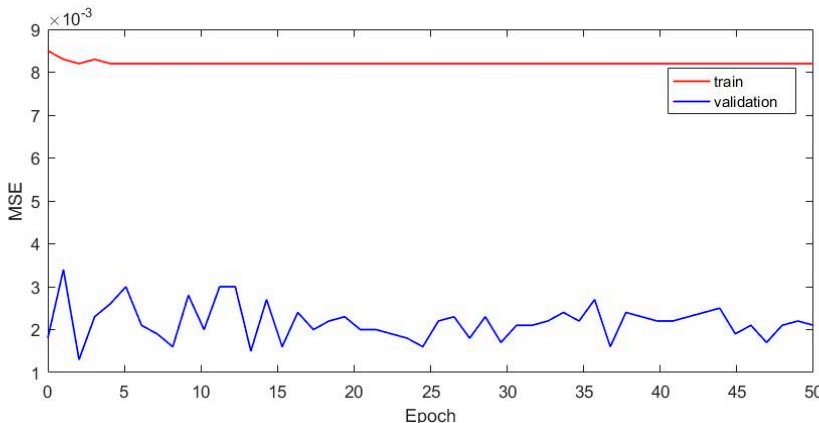

**Figure 11.** The MSE on the noise images corresponding to the training set and the validation set for VHWR.

However, as can be seen from Figures 5 and 6, the value of MSE obtained by the model with noise-free data is about $2.5 \times 10^{-4}$ when the noise variance is 0.01. This value is much lower than the MSE obtained by the model with noisy data, as shown in Figure 11. Given the poor performance of the models obtained from training with noisy data, we do not perform further testing experiments.

### 3.2. Vortex Quarter-Wave Retarder

#### 3.2.1. Noise-Free Data

Train

We examined algorithms on irradiance images when the light is modulated by a VQWR. In this case, the Stokes parameters are S1, S2 and S3. Here, the training and validation data sets are noise-free images. Figure 12 illustrates the MSE corresponding to various epochs, indicating that the MSE of the validation set is lower than that of the training set, which are all in the magnitude of $1 \times 10^{-3}$.

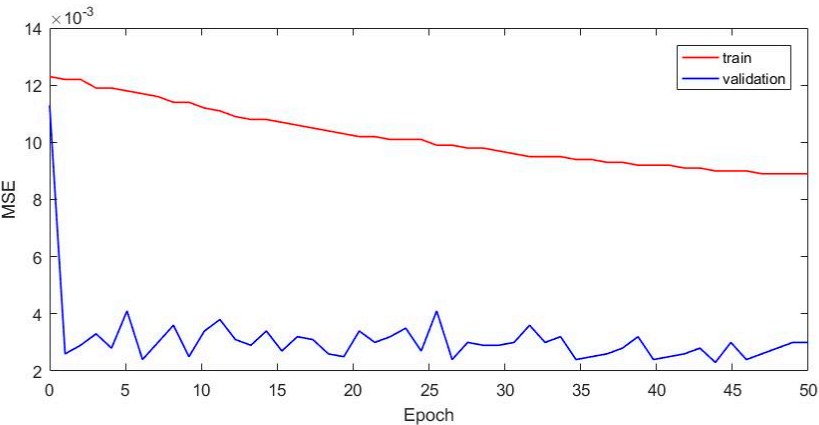

**Figure 12.** The MSE on the noise-free images corresponding to the training set and the validation set for VQWR.

Test

The training model with the smallest MSE in the training process was chosen as the best model for the test experiment.

To evaluate the robustness of the model to noise, 11 data sets are generated while the images are disturbed by Gaussian noise with zero mean and the variance ranging from 0 to 0.01 in steps of 0.001. Each data set contains 1000 images. We select one image from each data set as an example, and they are shown in Figure 13. The MSE of S1 estimated by ResNet-GAP, VGG and FAM is shown in Figure 14. The MSE of S2 and S3 is shown in Figures 15 and 16, respectively. The results show that, with the increase in the variance of Gaussian noise, the MSE of the Stokes parameters all increase gradually. In other words, all algorithms are sensitive to noise. However, compared to FAM, ResNet-GAP and VGG are more robust.

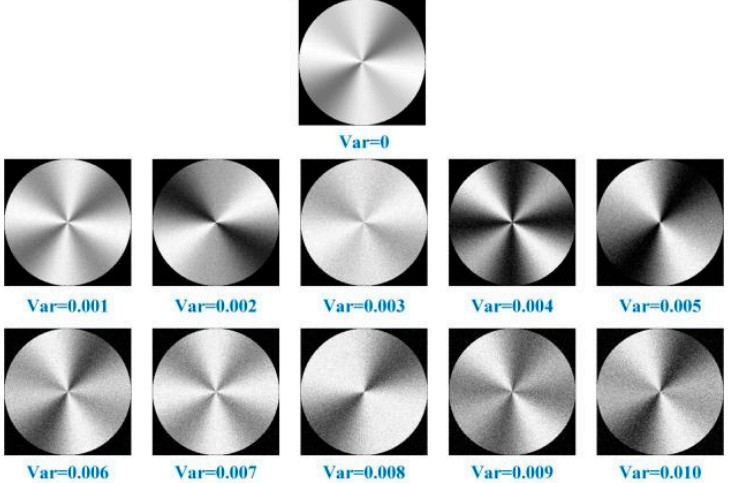

**Figure 13.** Images disturbed by different level of noise while the light is modulated by a VQWR.

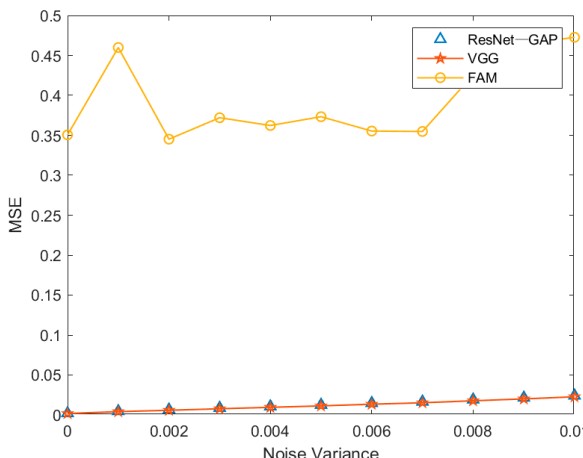

**Figure 14.** The MSE of S1 with respect to noise while the light is modulated by a VQWR.

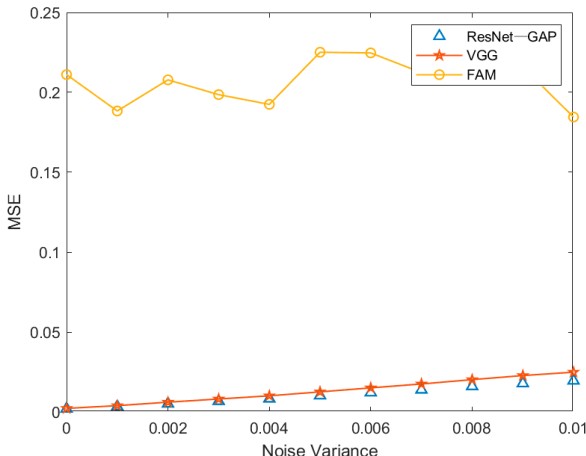

**Figure 15.** The MSE of S2 with respect to noise while the light is modulated by a VQWR.

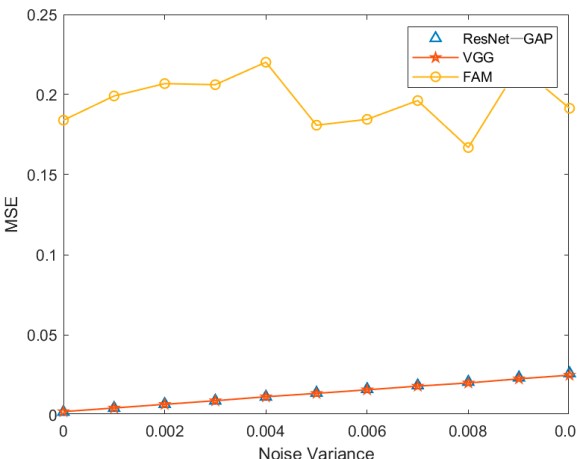

**Figure 16.** The MSE of S3 with respect to noise while the light is modulated by a VQWR.

Furthermore, the performances of different algorithms are tested on real data. A total of 37 real images are captured by our experimental system. The orientation of the fast axis of VQWR ranges from 0° to 180° in steps of 5°. The real images are shown in Figure 17. The results for S1, S2 and S3 shown in Figure 18 demonstrate the perfect performance of the algorithms. For ResNet-GAP, the maximum absolute error of S1 is 0.0794, and the average

value is 0.0172. For S2, the maximum absolute error is 0.0855, and the average error is 0.0073. For S3, the maximum absolute error is 0.0735, and the average error is 0.0193.

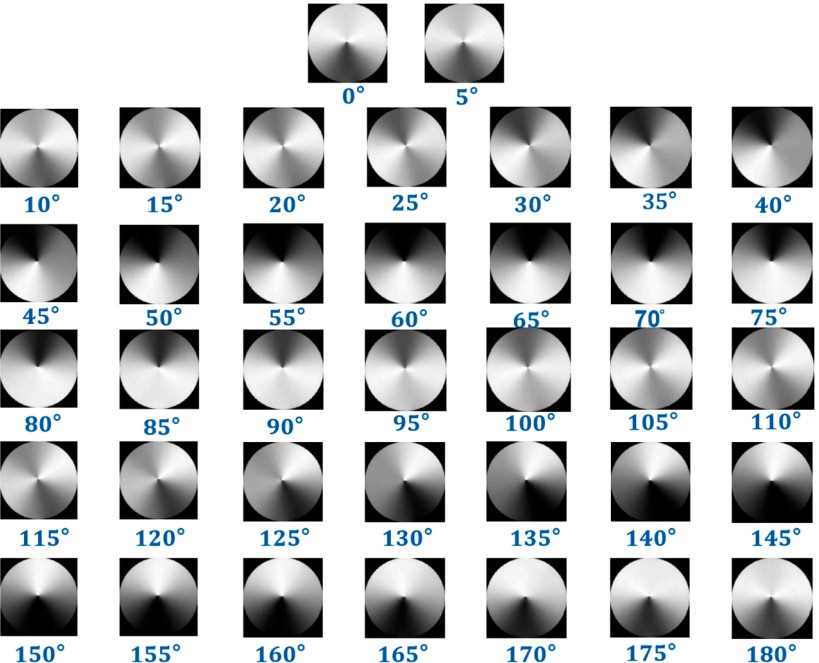

**Figure 17.** The real images when the fast axis of the VQWR is rotated.

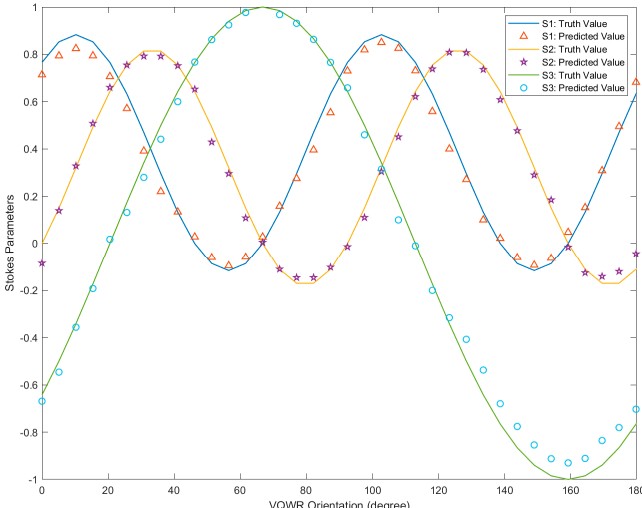

**Figure 18.** The Stokes parameters' real values and their evaluated values by ResNet-GAP.

The MSE of three algorithms are given in Table 3. As shown in Figures 19–21, the truth values of Stokes parameters are plotted against the predicted values of the three algorithms. They indicate that ResNet-GAP have the closest predicted value to the truth value and performed better than VGG and FAM.

The schematic diagrams of the Poincaré sphere are shown in Figures 22–25, where the blur curves represent the exact continuous distribution expressed in terms of the Stokes parameters, while the red asterisks represent the truth values and the predicted values obtained by ResNet–GAP, VGG and the FAM of the 37 real images respectively. According to the experimental results shown in these figures (from Figures 22–25), we can easily find that the measured results were well consistent with the predicted values in both the Stokes

curves and Poincaré sphere, and the fitted curves based on these discrete experimental results were also matched well with the theoretical predicted results.

**Table 3.** Performance of algorithms on real images for VQWR.

| Stokes Parameters | Network | MSE |
|---|---|---|
| S1 | ResNet-GAP | 0.0020 |
| S1 | FAM | 0.0120 |
| S1 | VGG | 0.0053 |
| S2 | ResNet-GAP | 0.0012 |
| S2 | FAM | 0.0032 |
| S2 | VGG | 0.0021 |
| S3 | ResNet-GAP | 0.0031 |
| S3 | FAM | 0.0046 |
| S3 | VGG | 0.0027 |

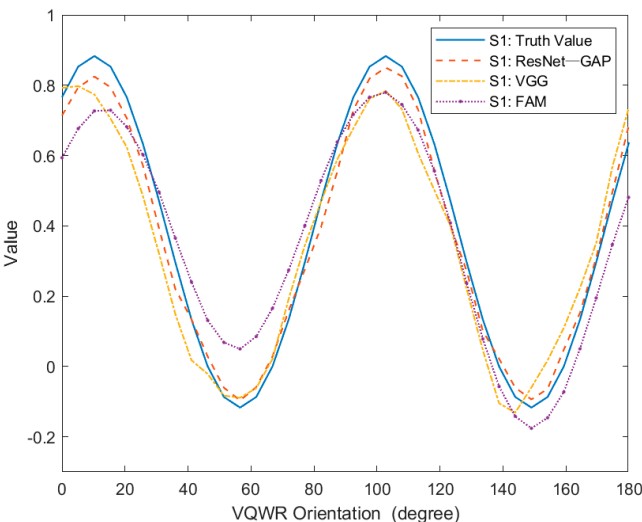

**Figure 19.** The values of S1 to different orientations of the fast axis for VQWR.

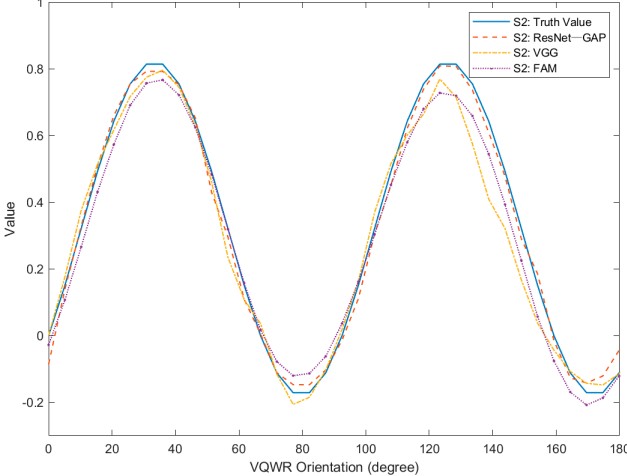

**Figure 20.** The values of S2 to different orientations of the fast axis for VQWR.

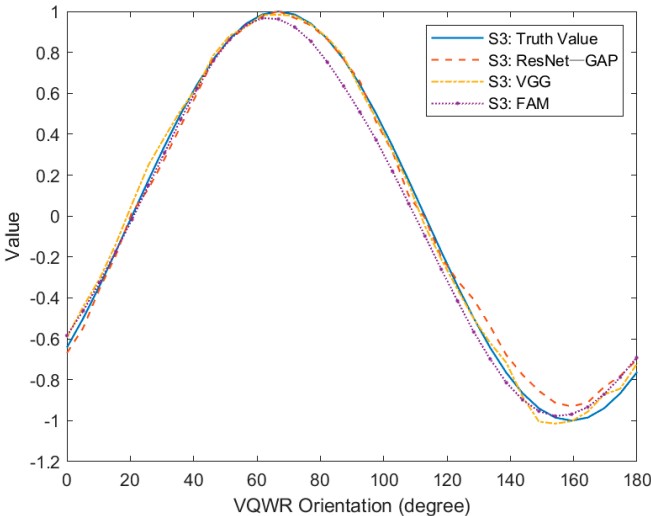

**Figure 21.** The values of S3 to different orientations of the fast axis for VQWR.

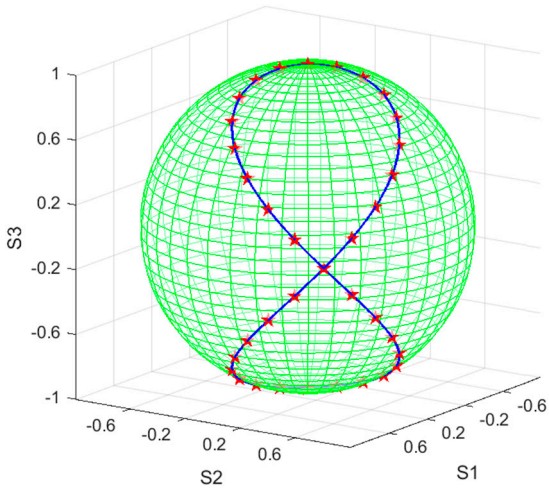

**Figure 22.** The schematic diagrams of the Poincaré sphere for the truth values.

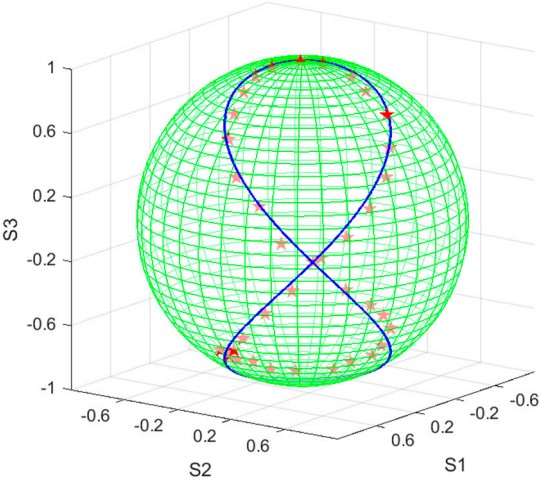

**Figure 23.** The schematic diagrams of the Poincaré sphere for the predicted values of ResNet-GAP.

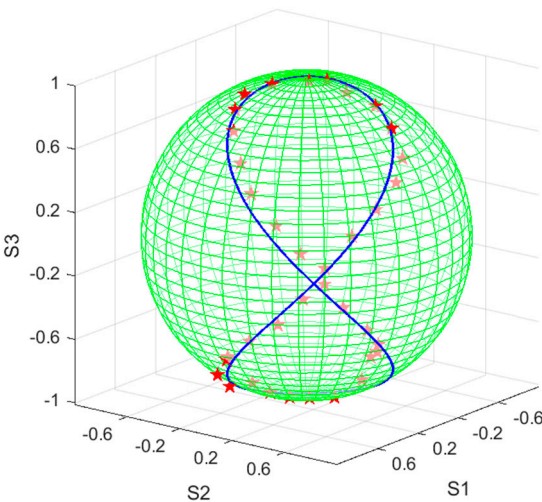

**Figure 24.** The schematic diagrams of the Poincaré sphere for the predicted values of VGG.

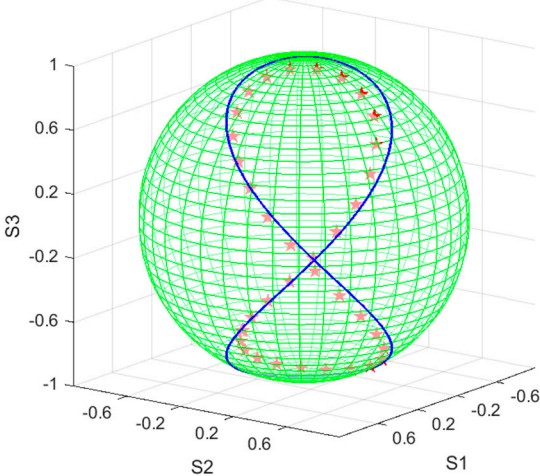

**Figure 25.** The schematic diagrams of the Poincaré sphere for the predicted values of FAM.

When 37 real images were tested, ResNet-GAP consumed only 0.02035 s, in comparison to VGG and FAM, which took 0.04162 s and 33.57 s, respectively. Obviously, the ResNet-GAP network is much faster than FAM and VGG. It is more suitable for real-time processing.

### 3.2.2. Noisy Data

Next, we use Gaussian noisy images with a zero mean and 0.01 variance as the training and validation data. All other parameters remain the same as in the VQWR without noise case. Figure 26 shows the MSE for different epochs. We observe that the MSE for the validation set is smaller than that of the training set. Both of them converge quickly, with MSE approximately in the magnitude of $1 \times 10^{-3}$.

However, the performance of the model obtained by training with noisy data is worse than that with noiseless data. Therefore, we do not perform further testing experiments.

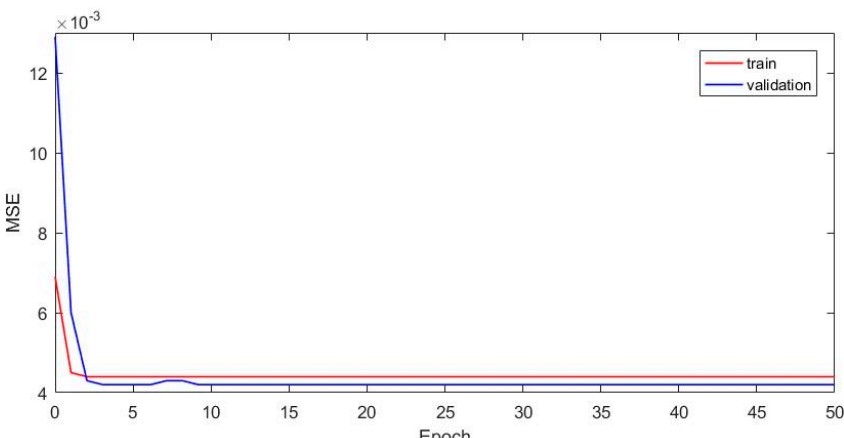

**Figure 26.** The MSE on the noise-free images corresponding to the training set and the validation set for VQWR.

## 4. Conclusions

In conclusion, we focused on how to effectively extract Stokes parameters of the light from the irradiance image in this paper. We proposed an improved Convolutional Neural Network which we call ResNet-GAP. The experiment results show that our proposed method can extract Stokes parameters effectively. We have tested on the synthetic and real data obtained from the VHWR and VQWR, respectively. Compared to VGG and FAM, the experiment results demonstrate that our method has outstanding performance with a smaller MSE and with a lower computational cost as well.

Although we have found the effectiveness of our proposed method, during our in-depth research, we also realized that some aspects are worth further research in the future. The first one is that our system uses a He-Ne laser with an operating wavelength of 632.8 nm; in future research, we will discuss the performance of ResNet-GAP under the width-wide wavelength case. Furthermore, we find that our network performs significantly better in testing real half-wavelength slice images than in testing real quarter-wavelength slice images under the criteria of MSE evaluation performance. We will further explore and explain this phenomenon in our subsequent studies. And, based on this, it is hopefully expected that we can obtain a better performance in testing real quarter-wave slice images.

**Author Contributions:** Conceptualization, W.W.; methodology, J.L. and C.G.; software, H.H.; writing—W.W., J.L., H.H. and C.G.; writing—review and editing, W.W. and J.S. All authors have read and agreed to the published version of the manuscript.

**Funding:** This research received no external funding.

**Data Availability Statement:** The data presented in this study are available on request from the corresponding author. The data are not publicly available, as the research group's polarization measurement is still being carried on, the later work will also rely on the current dataset.

**Conflicts of Interest:** The authors declare no conflict of interest.

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
