# Peer review of "An Improved CNN for Polarization Direction Measurement"

_electronics, doi:10.3390/electronics12173723_

Round 1

Reviewer 1 Report

1- On Line 32, this reviewer thinks computation and stability need to be further refined. The concerns they list, this reviewer does see as potential problems, but they have not addressed the complexity of those other systems in comparison to this methodology. What types of each (stability of optical materials to environmental elements? computation as in the computer needs are high (need a cluster or run)? or computation as in fitting a set of equations?) What are typical solutions in prior art? Why are those solutions not good enough now? 

2-This reviewer does not like the use of "and so on" in the introduction on line 33.  Leaving it open ended as that does not feel appropriate for a journal ready publication. A better way would be to not include that statement at all.

3- The second paragraph of the introduction is confusing because it is going through with this complex vortex retarder (a diffraction based optical element) but not ever actually discussing what diffraction is instead calling it a pattern of the light. This makes the later figures like Figure 3 and 8 extremely confusing because the dark areas shift and move. The use of the phase element could also allow for doing a method called division of aperture because you have shifting polarization within the field of view. I think the whole “why are you using the vortex retarder element” needs to be clarified.

4- By extractors in line 47 does the authors mean the line of equations that one solves like in Goldstein polarization book? There could be a better word for this term. Goldstein proposes similar simple stokes polarimeters with the same type of solving of intensity. You can easily do that without the complexity of the full system of a lot of the polarization systems discussed in the introduction. Something that is missing to this reviewer is with how much complexity a machine learning algorithm would have and how much time it would be to train how is that better than the simple systems in Goldstein? Is your method any more accurate? 

5- What wavelengths does this system operate on? Is it a one wavelength measurement? That is not detailed within this manuscript and would change the complexity of the system. 

6- In figure 1, where is the test component, you are changing within the system as described in Section 3? 

7- The authors have used two wave plates as images, but what would happen in terms of signal if there was a null (i.e. no data in the system). What is the remaining MSE in that case? 

8- How much accuracy of MSE would you need to have a working polarimetry system? Does this system meet that in the current configuration?

The English in this manuscript needs much improvement. There are spelling mistakes along with grammar issues. The language used in numerous sentences is awkward. This reviewer will run through issues found on the first page, alone.

Starting in the abstract, the second word polarization is misspelled. 

On line 10 of the abstract: "The classical imaging processing methods could not meet the increasing demand of practical applications" might be better worded as "Classical imaging processing methods could not meet the increasing demand of practical applications."

On line 15, the plural verb "are" needs to be changed to the singular "is". 

On line 26, the sentence beginning with "How to quickly capture the polarization state of light waves" is awkward. The beginning and end of the sentence are in essence saying the exact same thing.

On line 33, transform should be transforms because the verb needs to be the singular form. 

In Line 35 of the introduction, the word "doesn't" is used. Contractions, meaning a method of shorting commonly used words by joining them with an apostrophe, are not used in formal English publications. The proper words to use are "does not".  

Reviewer 2 Report

1) The problem statement is not clearly mentioned in the abstract? How do you say that image processing methods are not good? There is no literature based justification for this statement in the introduction section.

2) The introduction section lacks critical analysis

3) I do not find any improvement in the conventional CNN. Global average pooling is already available. How do you say that an improvement is made? Do you mean to say that just be including GPA, the efficiency is increased?

4) The methodology part is very weak.

5) The implementation part is missing. How do you fix the values?

6) Quantitative analysis is missing which is very important. How many date have you used? Will there be any change in the results if the data is changed?

7) Few graphs are given without any proper analysis on them. Why do you observe too many fluctuations in the graphs and could be the reason? Did you consider local minima during implementation?

8) A comparative analysis with other methods is missing. Without this, how can your results be validated and how can you justify your hypothesis?

Moderate changes required.

Reviewer 3 Report

This paper proposes an improved Convolutional Neural Network (CNN) for extracting Stokes parameters and polarization direction from irradiance images in spatial polarization modulation. The authors highlight the limitations of classical image processing methods in handling the increasing demands of practical applications due to poor computational efficiency. To address this issue, the proposed CNN utilizes residual blocks, different connected layers, and Global Average Pooling to enhance feature extraction and prevent overfitting. The algorithm is trained on a substantial dataset of synthetic images with and without noise and is evaluated based on mean square error (MSE) with the true values of normalized Stokes parameters.

Strengths:

- Effective utilization of Deep Learning: The paper leverages deep learning techniques to improve the efficiency and accuracy of polarization measurement. Deep learning has proven to be effective in various image processing tasks, and its application in polarization direction measurement is a promising approach.

- Adoption of ResNet and Global Average Pooling: The use of ResNet, a popular and powerful CNN architecture, along with Global Average Pooling to avoid overfitting issues and vanishing gradients, is a sound choice. These additions contribute to better generalization and performance of the model.

- Comprehensive Evaluation: The authors conduct extensive experiments on a large dataset, including synthetic images with and without noise. The use of MSE as the evaluation metric provides a quantitative measure of the algorithm's performance.

Weaknesses:

- Unsatisfactory English writing: The writing should be improved and extensive editing is recommended.

- Lack of Discussion on Latest Related Works: The paper lacks a comprehensive discussion and comparison with the latest related works in the field (e.g., "Learning to Super-resolve Dynamic Scenes for Neuromorphic Spike Camera" in AAAI 2023). This omission limits the reader's understanding of how the proposed "ResNet-GAP" algorithm compares to state-of-the-art methods.

- Limited Real-World Dataset: While the paper claims to address practical applications, it relies heavily on synthetic datasets. The algorithm's performance on real-world data should be evaluated to assess its robustness in real-life scenarios.

- Insufficient Discussion on Limitations: The paper lacks a thorough discussion of the limitations of the proposed algorithm. Addressing potential weaknesses would strengthen the paper's credibility and provide valuable insights for future research.

The English writing is generally understandable. However, there are a few areas where improvement could be made:

- Grammar and Sentence Structure: There are various grammatical errors and awkward sentence structures that impede the smooth reading experience. 

- Conciseness: The paper appears to be verbose and could benefit from more concise writing. Avoiding unnecessary repetitions and redundancies would help maintain the reader's interest.

Round 2

Reviewer 2 Report

It can be accepted now 

It’s fine